# The Effect of Adding Modified Chitosan on the Strength Properties of Bacterial Cellulose for Clinical Applications

**DOI:** 10.3390/polym13121995

**Published:** 2021-06-18

**Authors:** Anna Lipovka, Alexey Kharchenko, Andrey Dubovoy, Maxim Filipenko, Vyacheslav Stupak, Alexander Mayorov, Vladislav Fomenko, Pavel Geydt, Daniil Parshin

**Affiliations:** 1Lavrentyev Institute of Hydrodynamics of the Siberian Branch of the Russian Academy of Sciences, 630090 Novosibirsk, Russia; dubovoy@neuronsk.ru (A.D.); danilo.skiman@gmail.com (D.P.); 2Novosibirsk Research Institute of Traumatology and Orthopaedics n.a. Ya.L. Tsivyan, 630090 Novosibirsk, Russia; alexdok2000@gmail.com (A.K.); stupak@niito.ru (V.S.); 3Federal Neurosurgical Center, 630048 Novosibirsk, Russia; 4Institute of Chemical Biology and Fundamental Medicine of the Siberian Branch of the Russian Academy of Sciences, 630090 Novosibirsk, Russia; mlfilipenko@gmail.com; 5Institute of Laser Physics of the Russian Academy of Sciences, 630090 Novosibirsk, Russia; aleksander.mayorov@gmail.com; 6N.N. Vorozhtsov Novosibirsk Institute of Organic Chemistry of the Russian Academy of Sciences, 630090 Novosibirsk, Russia; vladislav@ngs.ru; 7Novosibirsk State University, 630090 Novosibirsk, Russia; p.geydt@nsu.ru

**Keywords:** tissue biomechanics, dura matter, bacterial nanocellulose, factor analysis, dura substitutes, chitosan gel, Novochizol™, vancomycin

## Abstract

Currently, several materials for the closure of the dura mater (DM) defects are known. However, the long-term results of their usage reveal a number of disadvantages. The use of antibiotics and chitosan is one of the major trends in solving the problems associated with infectious after-operational complications. This work compares the mechanical properties of samples of bacterial nanocellulose (BNC) impregnated with Novochizol™ and vancomycin with native BNC and preserved and native human DM. An assessment of the possibility of controling the mechanical properties of these materials by changing their thickness has been performed by statistical analysis methods. A total of 80 specimens of comparable samples were investigated. During the analysis, the results obtained, the factor of Novochizol™ addition has provided a statistically significant impact on the strength properties (Fisher Criteria *p*-value 0.00509 for stress and 0.00112 for deformation). Moreover, a stronger relationship between the thickness of the samples and their ultimate load was shown: $R^{2}=0.236$ for BNC + Novochizol™ + vancomycin, compared to $R^{2}=0.0405$ for native BNC. Using factor analysis, it was possible to show a significant effect of modified chitosan (Novochizol™) on the ultimate stress (*p*-value = 0.005).

## 1. Introduction

The substance of the brain and spinal cord is covered by three layers of connective tissue, collectively called the meninges: the soft (pia mater), the arachnoid membrane, and the dura (dura mater). The meninges contain blood vessels and are surrounded by cerebrospinal fluid (CSF). The outer layer of three layers of membrane, i.e., the dura mater, is an irregularly shaped, thick, whitish sheath of dense fibrous tissue with a large number of elastic fibers.

In the case of injuries, the oncological process of the brain and spinal cord, and during various neurosurgical interventions, the dura mater is dissected to provide access to the nerve structures located underneath. After the operation, the skull is sealed by suturing the dura mater. Often, for a number of reasons, this cannot be achieved, so during the postoperative period, liquorrhea (leakage of cerebrospinal fluid) develops in the area of the surgery, which leads to communication of the cranial cavity with the external environment. Such a postoperative complication is the most dangerous, as it often leads to the development of severe purulent complications from the central nervous system. With the development of such complications, intensive conservative therapy is needed in the form of prescribing antibacterial drugs, and repeated surgical interventions aimed at sealing the cranial cavity and spinal canal with the additional imposition of repeated sealed sutures on the dura mater are also required in some cases. Effective closure of the dura mater defect helps to minimize liquorrhea and promotes normal wound healing [1,2,3].

In addition, in violation of the dura mater integrity, fistulas and pseudomeningocele are formed (Figure 1). The latter is defined as a pathological extradural accumulation of cerebrospinal fluid in soft tissues communicating through a defect in the dura mater with the arachnoid space of the brain [4,5]. Figure 1 shows a cavity in the soft tissues of the cervical spine filled with cerebrospinal fluid (pseudomeningocele), which connects to the cerebrospinal fluid space of the spinal cord after removal of the extramedullary tumor and non-hermetic suturing of the dura mater. Plastic surgery on the dura mater is also necessary in cases of restoration of the lost part of the intrinsic dura mater associated with its invasion by a tumor, developed cerebral edema, elimination of cerebrospinal fluid fistulas, increase in the subdural space in Arnold–Chiari malformation, and myelomeningocele surgery [6].

From the available statistics of complications based on the clinical data, it can be concluded that it is necessary to repair dura mater defects using various types of implants since it reduces the number of postoperative complications [7,8,9]. For this, materials of various origins are used [10]:Autograft (fascia lata, fascia of the temporal muscle, pericranium et al. [11,12,13])Xenograft (collagen implants: DuraGen, Lyoplant^®^ et al. [7,14])Synthetic: absorbable (PGA, copolymer of L-lactic acid and epsilon-caprolactone, or copolymer of lactide and polydioxanone) and non-absorbable (ePTF, Polyesterurethane-Neuro Patch, polypropylene G-patch [15,16,17])Biopolymers (Chitosan, bacterial cellulose and et al. [18,19])

Sealing of the skull by dura mater repair using the body’s own tissues (autografts) is the most effective way [20]. However, the use of the patient’s autotissue for these purposes is limited by the laboriousness of the technique (splitting the dura mater according to the technique of N.N. Burdenko). In addition, the use of a portion of the hip fascia is an additional traumatic factor for the patient, fraught with lengthening the duration of the surgery and possible complications at the donor site [2]. The use of allografts (preserved cadaveric dura mater) is currently prohibited due to the possibility of transmission of infections and viruses, as well as difficulties in procurement and storage of material [21].

In the modern neurosurgery practice, there are a number of materials approved for use in the clinic for the closure of dura mater defects: DuraGen, DuraGen Plus, DuraGen Suturable collagen implants, Lyoplant^®^. All of them are pure collagen implants that are produced from lyophilized cattle pericardium. Collagen implants are the gold standard in neurosurgery for dura mater repair. However, all of these implants are imperfect and have their disadvantages.

Lyodura is a material from cadaveric dura mater produced by the German manufacturer B. Braun Melsungen AG and was the main source of the prion disease outbreak (Creutzfeld–Jakob disease). Laboratory studies have shown that standard methods of decontamination and sterilization may not be sufficient to completely eliminate prion contamination of surgical instruments after surgical treatment [22,23]. It is possible that collagen implants can transmit prions.

Synthetic implants often carry a high risk of infectious complications [24,25,26]. Synthetic grafts are often rigid and can cause inflammatory and foreign body reactions. These reactions can create inflammation of the surrounding tissue and brain, excessive production of fibrin during graft encapsulation, meningitis, graft rejection, scarring, and delayed bleeding, which often require reoperation [27].

Long-term results of using implants are not always completely satisfactory both for the patient and for the attending physician. Consequently, there is a need to find and develop materials (DM substitutes) that will be free from the drawbacks of existing implants and will have improved properties.

The general problems of using dura mater implants are as follows:Allogeneic tissues: xenograft tissue can cause its rejection [28].Dura mater and surrounding tissue adhesion: after the restoration, the dura mater implants have a different degree of adhesion, mainly associated with the inflammatory response, physical and chemical properties of the material. The lower the content of protein and fat in the material, the lower the degree of adhesion [29].Development of aseptic inflammation. In addition, the use of allogeneic and xenogenic materials can lead to the spread of pathogens among humans and animals, prions and viruses [30].Bleeding: incipient granulation tissue that regenerates and covers the graft material can cause bleeding. There may be a gap between the material and the elaborated network of neocapillaries covering the matrix. The capillaries are fragile, bleeding can occur when the implant is mixed, and a subdural hematoma can form in the cranial cavity [31].Development of liquorrhea as a result of the lack of reliable sealing of the dura mater defect [32].The occurrence of epileptic seizures, as a result of the development of meningeal adhesions [33].

The ideal material for plastics of the dura mater has not yet been created, but the requirements for the design of the material to be used to close the dura mater defect are that it should not induce an immunological or inflammatory response, should not be neurotoxic, carcinogenic, should provide a hermetic closure, and retain its shape after use and stay durable. An ideal DM substitute should not pose a risk of transmission of viral and prion infections. The implant should be capable of being stored for a long time, retain its valuable properties, and have a low cost.

In our opinion, a promising direction is the use of implants made of bacterial nanocellulose, since this material meets the above-mentioned requirements.

In neurosurgery, it is important that new materials for dura mater restoration not only perform a barrier function but also have necessary mechanical strength. This is because increased intracranial pressure can rupture a new dura mater implant after decompression craniotomy.

In the work of Dutta P.K. et al. [34], the effect of the selected bacterial strain on the growth of the BNC polymer was investigated. The authors have shown a correlation between the ultimate strength and the concentration of the polymer in the hydrogel, while the ultimate deformation did not correlate with the concentration of the polymer. The choice of a specific strain of bacteria for the production of BNC material, as can be seen from their work, is also important.

In the article by Kizmazoglu C. et al. [35], it was shown that mechanical properties of the DM substitutes vary significantly and often greatly differ from the properties of DM itself. Nevertheless, the studied DM substitutes have shown their effectiveness. Moreover, the tensile strength of these substitutes was measured to be exceeding the DM strength for some materials, while it was lower than DM for others. After comparing the results from similar studies, it can be noted that the difference in strength in DM and its substitute can be significant, up to 10 times.

Such differences can be explained not only by different approaches to measuring deformation but also by different approaches to material fixing. For example, in the works [35,36], fixing was used in clamps with a uniform contact surface. On the one hand, this simplifies the manufacture of such clamps, but on the other hand, it requires significant pressure on the sample to achieve a larger contact surface and, therefore, a greater frictional force to hold the sample. This, as seen in [36], dramatically affects the boundary deformation of the sample. In our approach, a developed technique of fixing the sample was used, which increased the area of the contact surface of the clamps and the sample due to the irregularities of the clamps themselves (due to the presence of special grooves and protrusions).

Chitosan has important properties such as biocompatibility, biodegradability, hydrophilicity, non-toxicity, high bioavailability, favorable water permeability, the ability to form films, gels, and nanoparticles. Chitosan nanoparticles (Novochizol™) were included in composite material for plastics of DM defects. Novochizol™ has a positive charge, which allows it to interact well with various types of molecules. It is believed that this positive charge is responsible for the antimicrobial activity of chitosan through interaction with negatively charged cell membranes of microorganisms [37]. By adding an antibiotic to composite material, we enhance the antibacterial effect of the composite material. If one has a native bacterial pulp antibiotic without chitosan, then the excretion of the antibiotic occurs within one hour. Novochizol™ forming a film gives a slow release of glycopeptide vancomycin. In contrast to chitosan, which is a linear polymer, Novochizol™ has a globular, near-spherical shape, owing to intramolecular cross-linking. Such a molecular design confers several advantages to Novochizol™ over chitosan, including the functioning as an active ingredients carrier (Table 1). It should also be noted that due to the fact that Novochizol™ is based on nanospheres, the diffusion rate is much higher than for ordinary linear chitosan, which is especially important when these materials are impregnated.

These properties of chitosan (see Figure 2) are ideal for use in a system of hydrophilic antibiotics with slow release, which is extremely important for the prevention of infectious complications during the closure of dura mater defects. In addition, vancomycin is used to enhance the antibacterial properties of chitosan. However, when native BNC is impregnated without Novochizol™, vancomycin is rapidly washed out [38].

Despite the fact that the mechanical properties of native cellulose have been studied for a long time in the literature, and the results of the effect of nanoadditives on the mechanical properties of DM substitutes have been reported before (Tutopatch^®^), the controlling of properties by material thickness has not been published previously. Remarkably, it has paramount importance for clinical patient-specific applications.

The purpose of this work is to compare the mechanical properties of samples of BNC impregnated with Novochizol™ and vancomycin with native BNC, cadaveric DM (preserved with formalin) and native human DM to assess the possibility of controlling the mechanical properties of the material by changing its thickness and also to evaluate how the additives affect the mechanical properties of the composite.

## 2. Materials and Methods

### 2.1. Ethical Protocol and Transportation

According to the established protocol, fresh dura mater samples were obtained during microsurgical treatment of patients with cerebral vascular pathologies. With unexpressed cerebral edema, the excess part of the dura mater was excised for research. Our clinical approach to tissue harvesting assumes the maximum safety of tissue extraction for the patient; therefore, only a small fragment of the dura mater was taken, which in any case would have been excised. This part of the study was carried out jointly with the Federal Neurosurgical Center of Novosibirsk. Harvested tissues were preserved with 0.9% saline at +2–5 ${}^{\circ }$C during transportation and storage until the experiment (12–48 h, which is a standard period in the literature.).

Cadaveric preserved DM tissues were selected in the laboratory for the preparation and preservation of tissues at the Novosibirsk Research Institute of Traumatology and Orthopedics n.a. Tsivyan, according to the local ethical protocol.

### 2.2. Methods for the Synthesis Native BNC

Native BNC was obtained by standard cultivation in the medium of the bacterial strain Komagataeibacter Xylinus JCM 7644 at the Institute of Chemical Biology and Fundamental Medicine SB RAS.

### 2.3. Methods for the Synthesis of Modified BNC

The BNC + N + V samples were prepared under special conditions. The native BNC, grown at the Institute of Chemical Biology and Fundamental Medicine of the SB RAS, was transferred to the Vorozhtsov Institute of Organic Chemistry SB RAS for treating with Novochizol™ (Registered International trademark Novochizol No. 1540749, and in U.S. Patent and Trademark Office No. 6297647). Chitosan nanospheres—Novochizol™ were provided by NOVOCHIZOL SA (Monthey, Switzerland, www.novochizol.ch, accessed on 23 March 2021). The degree of deacetylation was no less than 90%, and the mass is 500 kDa. Novochizol™ aqueous solutions were obtained by dissolving succinic acid (500 mg per 100 mL sterile water), gradually adding Novochizol™ (1000 mg per 100 mL succinic acid solution) under sonication, and sonicating the mixture for one hour, using model UZTA-0.4/22-OM sonicator (U-sonic, Biysk, Russia) at maximum power. Sterile water was added to compensate for evaporation caused by the prolonged sonication. The solution was filter-sterilized using 0.45 µm apyrogenic acetate cellulose filters (Minisart^®^, Sartorius Stedim Biotech Göttingen, Germany). Thereafter, 1000 mg of vancomycin hydrochloride was added to the resulting solution, and the solution was subjected to the same ultrasonic treatment for 5 min. The solution was filter-sterilized again using 0.45 µm apyrogenic acetate cellulose filters (Minisart^®^, Sartorius Stedim Biotech Göttingen, Germany), then stored as a 1% stock at +4 ${}^{\circ }$C and used within one week.

Samples of BC were individually immersed in a Novochizol™ solution in a plastic tube (50 mL) at a ratio of sample volume to Novochizol™ of 1:10. The samples in Novochizol™ were then treated with an ultrasonic bath at +37 °C for 10 min. After that, the samples impregnated with Novochizol™ were neutralized with 1M aqueous ammonia to pH7, washed with sterile water and stored at +4 °C until implantation. The fundamental differences of Novochizol™ from chitosan are indicated in Appendix A.

### 2.4. Elemental Analysis of BNC Samples

Elemental analysis of unimplanted samples was performed to identify possible damage as a result of their coating and sonication. Elemental analysis was carried out on an instrument “Carlo Erba Strumentazione, Elemental Analyzer-Mod. 1106, Milano, Italy”.

### 2.5. Atomic Force Microscopy

Atomic Force Microscopy (AFM) measurements were done in Semicontact Tapping mode on the Solver Next SPM station (NT-MDT, Russia) in standard atmospheric room conditions (RH 30%, temperature 25 ${}^{\circ }$C). Semisoft AFM probes NSG01 (NT-MDT, Russia) with resonant frequency 138 kHz and nominal tip radius 10 nm were used for the studies with the optimized setpoint force. All recordings were done with a scan rate of 0.5 Hz and resolution 512 × 512 points. Further data treatment involved only removal of inclination without additional data filtration.

### 2.6. Thickness Measurement and Material Cutting

The thickness of the investigated samples was preliminarily measured in the laboratory of the Institute of Laser Physics SB RAS. The thickness of 4 groups of materials was measured by a highly sensitive LVDT sensor, rigidly connected on one axis with a precision wheel 2 mm wide. The wheel evenly moves along the tissue flap from left to right and from top to bottom, transmitting vertical vibrations to the LVDT sensor, the signal from which is sent to a personal computer, where the developed software directly displays the thickness of the biological material. Before the cutting, a “thickness map” of the tissue flap with an accuracy of ±10 μm is created, where the selected thickness range corresponds to the color specified by the operator (Figure 3). To conduct a reliable experiment, the same material thickness in the middle part of the “dog-bone” shape was chosen.

For cutting out samples from the sheet, a laboratory installation for cutting out biological tissue “Melaz Cardio” (LLC “Lasarus”) was used. The cutting was carried out in a continuous mode using a CO_2_ laser, the power of which can reach 40 W, which was regulated depending on the material. Cutting speed was 600 mm/min, while power was 33%.

### 2.7. Mechanical Test Protocol

All mechanical tests were carried out at the Lavrentyev Institute of Hydrodynamics SB RAS on a universal tensile testing machine INSTRON 5944 with a thermostatic biobath. During the test, each sample was subjected to cyclic loading with a starting displacement of 0.25 mm at the first cycle, with a displacement step of 0.25 mm and a crosshead speed of 2 mm/min. After delivery to the laboratory, the thickness of the sample was measured (for DM). Then the sample was fixed in an Instron 5944 tensile machine, and a series of experiments were performed (Figure 4b,c). A dog-bone shape was cut from the cadaveric material with the same parameters as for BNC and the BNC + N + V composite material. For fresh dura mater samples obtained during neurosurgical interventions, a rectangular sample was used for testing, mainly due to the small size and irregular shape of the original samples, as well as to simplify data processing. A videoextensometer (lens from Fujifilm) was used to measure local strain in the specimens, when their size allowed. Small bright plastic marks (1.5 mm in diameter) were attached to the sample with the water-proof glue for the extensometer to capture gauge length (Figure 4) during the loading. The focal length of the video extensometer objective is 16 mm.

The video extensometer used to measure local deformation allows minimizing the boundary effects, which are prone to happen in the case of measuring only the crosshead strain. The known parasitic effects caused by the clamps, such as slippage of the sample, were considered. If the clamps were too loose, then the part of material under them could extend, as well as the part outside the clamps, or even slip out [39]. Uneven deformation of the material (demonstrated, e.g., in [36] between the clamps also motivated the acquisition of deformation data in two ways: by considering edge effects and minimizing them. In addition, damage to the material could be introduced while fastening the sample. This established the zones of already partly ruptured material, which negatively affected the accuracy of the obtained experimental data. All of that emphasizes the need to use other means of measurement apart from the crosshead strain.

While carrying out the experiment, such a well-known phenomenon for biological tissues as preconditioning [40] was taken into account. This technique was applied for the initial stages (1st–5th stages of elongation, depending on the sample), and during the next stages, the influence of this condition was not noticed. During this study, it was established that was no need to perform more than two preconditioning cycles. For each stage of the experiment, the specimen’s initial elongation was the same, i.e., the machine’s clamps returned to the original program-defined position after the completion of each stage of the loading. During the experiment, the sample was positioned in the sodium chloride solution heated to the human body temperature. In each experiment, the sample lost its elasticity. For each sample, its loading was performed until it became separated into two disconnected segments (or a visible discontinuity of the sample appeared).

### 2.8. Statistical Analysis

To analyze the results, the following software were used: the MS Office 2016 (MS Excel with custom extensions) package licensed by LIH SB RAS and free software environment R. Truncated samples relative to the original ones (10% of the highest and lowest results were discarded) were formed, which is determined by outliers in the data that are observed not only in this work but also in the literature [41].

## 3. Results

### 3.1. Results of AFM of BNC

The analysis of AFM topography images provided observation of separate and bundled cellulose nanofibrils with the diameter in the range of 30–50 nm (see Figure 5a), which were evenly spread around the surface (as seen on Figure 5b). These observations were supported by Scanning Electron Microscopy data (not shown) and verified by the splitter-fiber structure of the sample.

### 3.2. Elemental Analysis of Novochizol™

The following element ratio was established for Novochizol™ (see Table 2).

That elemental analysis of BNC + N sample shows that Novochizol™ and cellulose are in approximately equal weight proportions.

### 3.3. Characteristics of a Sample of Tested Specimens

A total of 43 experiments were carried out on uniaxial mechanical loading of BNC specimens and 22 BNC + N + V specimens. In addition, tissues of the dura mater (4 samples) of healthy patients and samples of the cadaveric dura mater (13 samples) were examined using the same technique. After processing the test data, it was decided to present the results of the measurements made with the data taken from the traverse displacement of the stretching machine (Figure 6a) and the data of the video extensometer (Figure 6b). It is visible that the data on the ultimate deformation of the samples can differ significantly.

To analyze the mechanical properties of such materials, it is important to analyze not only their ultimate stress indices but also the elastic modulus at small deformations, since exactly the small deformations of the DM are the most physiological type of deformations of such a material. Thus, the atypical behavior of the material in the region of small deformations with its adequate limiting mechanical characteristics may indicate an unsatisfactory quality of the material.

### 3.4. Statistical Analysis of Mechanical Test Results

In the course of the study, it was shown that the ultimate stress of the new composite is similar and slightly exceeds the ultimate stress of the material grown according to the standard technique: 0.75 vs. 0.58 MPa (+29.31%) (Table 3).

The ultimate deformation of the cadaveric dura mater is in accordance with Figure 7a 1.001, the ultimate deformation of BNC is 1.002, and the ultimate deformation of the BNC + N + V composite is 1.125. Thus, the ultimate deformation of the new BNC + N + V composite is 12.4% higher than the cadaveric dura mater.

In the course of the study, in accordance with Figure 7b, it was shown that the ultimate stress of the new composite (BNC + N + V) slightly exceeds the limiting stress of the native BNC material grown according to the standard method: 0.74 and 0.54 MPa. The ultimate stress of the new composite (BNC + N + V) is 36.54% higher than the limiting stress of the native BNC. The limiting stress of preserved cadaveric DM (2.387 MPa) is 336.6% higher than that of native BNC. The results of the ultimate stress were also compared with the results of the DM and Tutopatch^®^ presented in [35]. Tutopatch^®^ (Tutogen Medical GmbH, Neunkirchen am Brand, Germany), which is produced from bovine pericardium, is xenogeneic and exposed to the Tutoplast process [43]). The Tutoplast process is chemical sterilization, which increases the strength of bovine pericardium to enzymatic breakdown and decreases its antigenicity.

Statistical analysis of the dependency between the thickness of the samples and their ultimate values of stress and strain, as well as the Young’s modulus at small deformations, showed that no significant differences between the two types of material (BNC vs. BNC + N + V) can be observed from their Young’s modulus. Regarding the relationship between ultimate stress and deformation, the relationship for the standard BNC cultivation method is more linear (R=0.306), at the same time, this only indicates that this relationship is less linear for BNC + N + V material (Figure 8). Analyzing the relationship between the thickness of the material and its ultimate stress, it can be seen (Figure 9) that this relationship is more linear for the new material ($R^{2}=0.235362$ for BNC + N + V, compared to $R^{2}=0.040461$ for native BNC), and in this case, this indicates more predictable strength characteristics of the material during its growth. Therefore, with the thickness–stress relationship being linear, it is enough to carry out the measurements of the material thickness at the same definite time intervals rather than building a complex polynomial or recursive model for measuring the thickness of the grown material, which is an undoubted technological advantage of such a material.

An analysis of variance was also performed for the values of ultimate stress and deformation with respect to such a factor as the presence of Novochizol™ in the material (Figure 10). The results showed a significant effect on both parameters (*p*-value 0.005 for stress and 0.001 for deformation). That is, the impregnation of the material with Novochizol™ statistically significantly increases both the strength and elasticity of the material.

## 4. Discussion

Bacterial Nanocellulose (BNC), synthesized by the strain bacterium Komagataeibacter xylinus JCM 7644, consists of a biogenic structure of nanofibers formed by self-assembly. BNC has a higher water retention capacity, excellent biocompatibility, a high degree of crystallinity and, therefore, a high tensile strength and fine mesh compared to pure natural biodegradable polymers such as collagen, chitin and gelatin [44,45,46,47].

BNC possesses high mechanical properties, which are required in most cases when the material is used as a base in the tissue engineering. With a reticular structure and very small pore sizes, protofibrils of bacterial nanocellulose intertwine to form a large surface area. The fibrous structure of BNC consists of a three-dimensional network of nanofibrils connected by intrafibrillar hydrogen bonds, which allows it to maintain a constantly wet state of the hydrogel, as well as the high strength of BNC [47,48]. In the last decade, BNC has been widely used to create biocompatible prostheses in human tissues [19,47]. For materials used as dura mater transplants, especially those with high hydrophilicity, very important properties are the deposition of drugs and a decrease in toxicity by reducing leaching and, as a consequence, local action and achieving prolonged drug release.

For this, a variety of polymer sustained-release drugs with different physical properties have been developed. Such formulations have been shown to be effective in increasing the release time when relatively hydrophobic and water-insoluble drugs are used. However, there is still a need for new compositions and methods that could reduce the diffusion of the drug and eliminate the “explosion” effect for drugs that are highly soluble in water.

Currently, the world literature is actively highlighting the prospect of using a natural polymer of chitosan [49,50,51,52]. It is noteworthy that the entire 62nd issue of Advanced Drug Delivery Reviews (2010) was devoted to this material. Chitosan, which is a mucopolysaccharide, resembles the structure of the polymer lining the intima of blood vessels. It is not surprising that it has complete biocompatibility with human tissues. Its low toxicity, ability to enhance regenerative processes during wound healing, and biodegradability of chitosan materials are of particular interest for medical use [18,53,54].

Biodegradation of Novochizol™ will lead to an increase in the duration of the release of antibacterial and antimycotic drugs, which prolongs their effect in the tissues adjacent to the composite material. Due to the use of the BNC + N + V composite, the number of bacterial and fungal complications during neurosurgical interventions will decrease.

In this regard, it should be noted that, given the degradation over time of BNC with medicinal coatings (additives), namely, the release of the medicinal agent, the change in mechanical properties is of great interest. In Reference [55], structural changes in the polymer are described as a function of changes in the concentration of the drug additive.

In other studies, only the mechanical properties of BNC are considered, from the basic uniaxial test protocol [56] to the advanced measurement techniques that allow characterizing the material in sufficient detail (to determine not only Young’s modulus but also Poisson’s ratio by performing a compression test) [57].

With all the advantages of BNC as a transport and separation barrier for the human brain, the main problem during its growth is the unevenness of tissue thickness, which, during decompression craniectomy and with the emerging cerebral edema, going beyond the trepanation window, can lead to the occurrence of hernias and ruptures in the BNC film after its implantation. Therefore, the strength properties of materials for meringoplasty are very important.

This question poses an additional challenge, since, as our results show, regulating the thickness of bacterial cellulose allows one to regulate its strength properties. However, such a dependence, which is also shown in the Results section, is clearly non-linear, which makes it difficult to grow this composite.

In terms of mechanical properties, plastic materials used to replace DM defects should be as close as possible to the properties of natural DM. The mechanical properties of the material (implants) include ultimate stresses and deformation, which are described in [35,36], as well as strength characteristics in the area of small deformations (Young’s modulus, dependence of deformation on stress), which was studied.

The most common method for studying the strength properties of DM is the study of the mechanical properties of cadaveric specimens. This is usually due to ethical standards (inability to take a DM sample from a healthy person within the framework of ethical protocols) and established clinical protocols. However, the use of formalin significantly changes the strength properties of the bacterial cellulose material [58]. In turn, obtaining fresh samples of dura mater is difficult since, in the general case, excision of dura mater is not provided for by the protocols and can be performed in exceptional cases.

The mechanical properties of BNC were studied both to recognize the strength properties of this material for the purposes of implantation instead of DM [35] and for other applications in the sense of implantation, for example, in [59]. Undoubtedly, the test method is determined by the application of the target material. In this case, as a DM substitute, the implanted BNC is experiencing stretching stresses; therefore, it seems natural to use an uniaxial mechanical test [60,61]. In this paper, the most advanced method of uniaxial testing is applied: the use of the most sensitive dynamometer (10N Load Cell), the use of a thermostatic bath and the use of a video extensometer to fix the truth, and calculating relative deformations. In addition, we used dog-bone-shaped samples for the most correct localization of the sample break zone during a mechanical experiment [62]. To provide the possibility of using the results of mechanical tests by other researchers, we provide access to the results of the experiments in the Appendix A.

The limitations of this work include a small number of dura mater samples from healthy patients. Although, it is worth noting that this indicator is rather an overview at this stage of the study and does not affect the conclusions drawn. In addition, during neurosurgical intervention, dura mater samples usually cannot be removed, unlike cerebral aneurysm tissues [63], which can be excised, and sometimes must be removed, e.g., if there is a mass effect on the patient’s brain. In addition, recently, the technique of multi-axis experiments has been developed, despite the fact that the advantage of multi-axis tests [64] is hampered by the complexity of the interpretation of such tests [65]. It is impossible to disregard the usefulness of such tests; therefore, similar tests should be performed in the future.

## 5. Conclusions

In this work, a comparison of the strength properties of two cohorts of a bacterial nanocellulose polymer (native and impregnated with Novochizol™ + vancomycin) was conducted and showed the effect of the additives on the strength characteristics (the additive plays the role of a reinforcement factor). The strength characteristics of the BNC + N + V composite polymer have been compared with the DM material (fresh and cadaveric), and similar characteristics of the ultimate stress for the BNC + N + V and DM materials were shown, which points to the adequacy of the considered BNC options. The results obtained shed light on the change in the strength characteristics upon addition of Novochizol™ and on the possibility of controlling the strength properties of the polymer using the polymer thickness. In the future, it is planned to carry out punch tests on a hard surface. Particularly, the anticipated study of the curved surface, analogous to the material near the skull, seems promising for the development of real applications.

## Figures and Tables

**Figure 1 polymers-13-01995-f001:**
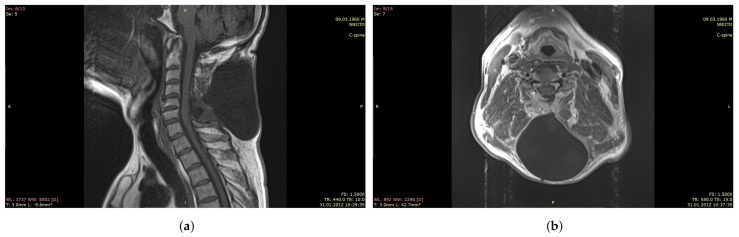
MRI of the cervical spine after removal of the spinal cord tumor and leaky suturing of the dura mater with the pseudomeningocele formation. (**a**): Sagittal plane, (**b**): Horizontal plane.

**Figure 2 polymers-13-01995-f002:**
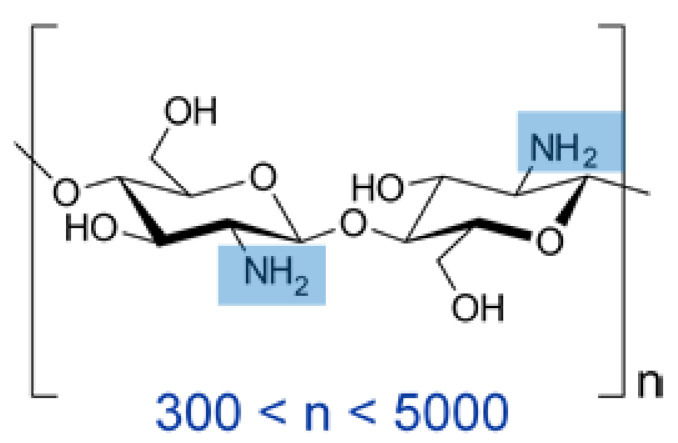
Spatial structure of the chitosan molecule—schematically.

**Figure 3 polymers-13-01995-f003:**
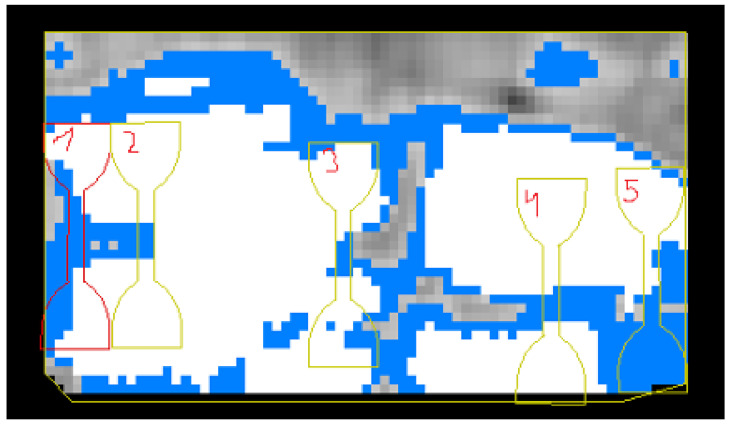
The image was obtained on a laser thickness gauge with markings for subsequent cutting of tissue with a laser. The white zone corresponds to a thickness of 1.0 mm, the blue zone—0.7 mm, the gray zone—0.3 mm.

**Figure 4 polymers-13-01995-f004:**
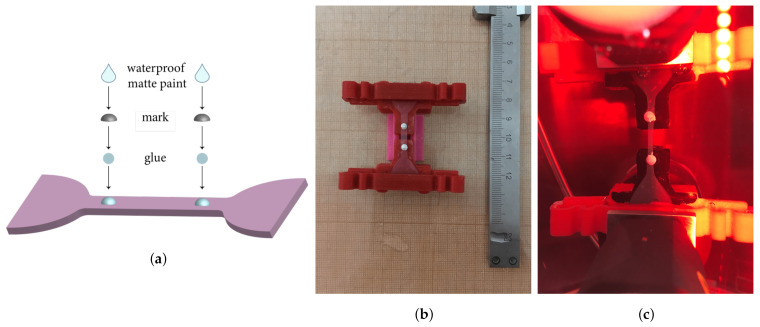
(**a**)—The technique of attaching marks for extensometer, (**b**)—preparation of the specimens, (**c**)—specimen in the tensile machine.

**Figure 5 polymers-13-01995-f005:**
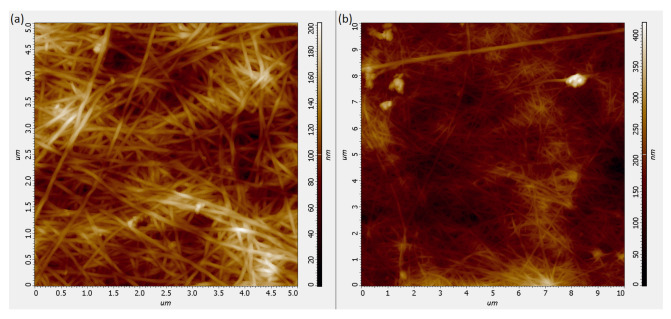
Topography of the BNC sample obtained by Atomic Force Microscopy for 5 × 5 μm area (**a**) and 10 × 10 μm area (**b**).

**Figure 6 polymers-13-01995-f006:**
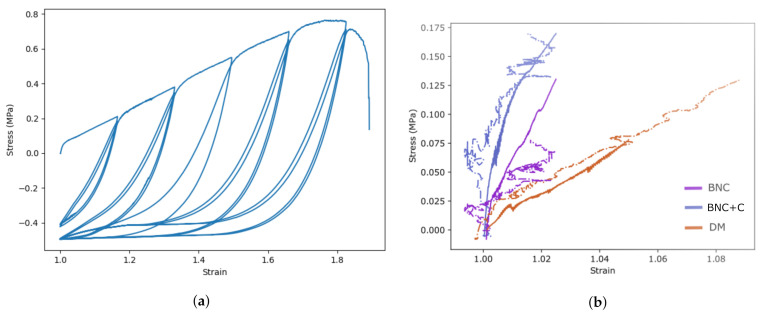
(**a**)—The course of the tensile test, (**b**)—first stage of the testing for different materials. The continuous line denotes the crosshead strain, and the dotted line denotes the video extensometer strain.

**Figure 7 polymers-13-01995-f007:**
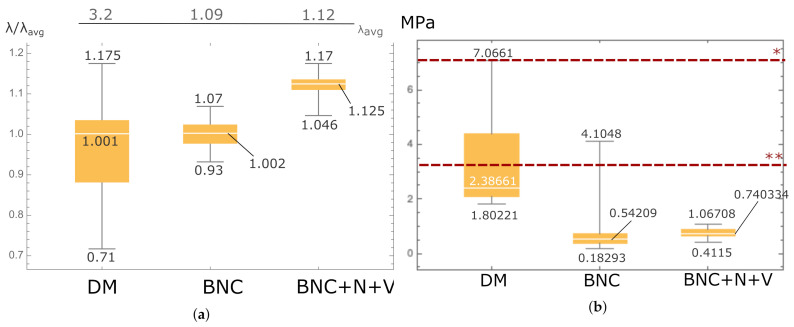
Boxplot for values of ultimate strain (**a**) and stress (**b**). The lines (*) and (**) correspond to data of the same value from [35] of DM and Tutopatch^®^ material. Reprinted from [42].

**Figure 8 polymers-13-01995-f008:**
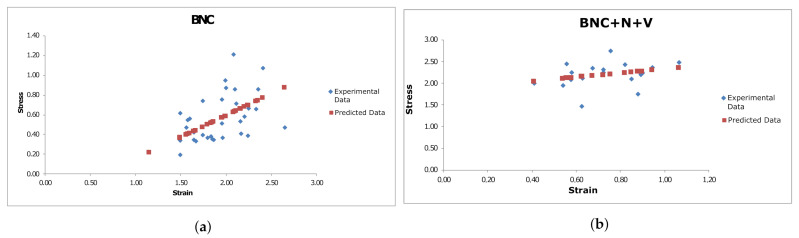
Linear regression for BNC (**a**) and BNC + N + V (**b**) samples for stress-strain relationship. Reprinted from [42].

**Figure 9 polymers-13-01995-f009:**
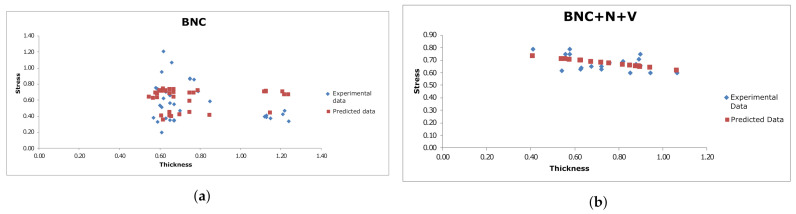
Linear regression for BNC (**a**) and BNC + N + V (**b**) samples for the stress–thickness relationship. Each blue dot represents one sample. The red dots represent the result predicted by the linear regression. Reprinted from [42].

**Figure 10 polymers-13-01995-f010:**
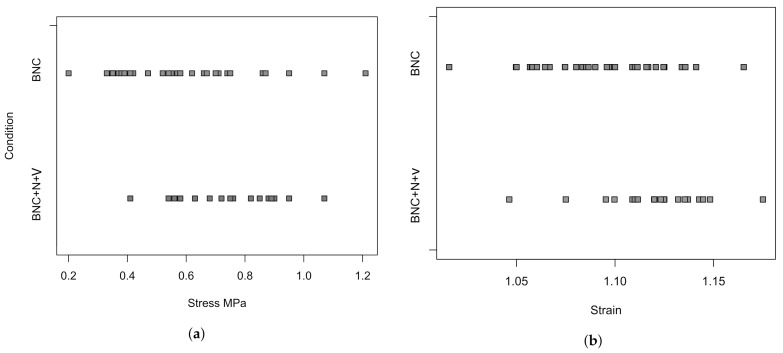
An illustration for analysis of variance. (**a**): the values of the ultimate stress for material with and without chitosan, (**b**): for the ultimate deformation. (Each point represents one sample).

**Table 1 polymers-13-01995-t001:** Characteristics of Novochizol™ of importance in plant treatment, in comparison with chitosan.

Characteristic	Chitosan	Novochizol™
Solubility (pH < 6)	Yes	Yes
Solubility (pH > 6)	No	Yes (dispersion)
Viscosity	High	Low
Biodegradability	Fast	Slow
Chemical stability	Low	High
Frost resistance	No	Yes
Physical states	Modifiable only	Modifiable by
	through chemical	changing the
	reactions with	degree on intramolecular
	other compounds	cross linking

**Table 2 polymers-13-01995-t002:** Results of elemental analysis of BNC, BNC + N, BNC + N + V.

Sample Type	C, %	H, %	N, %
BNC	39.73	7.11	7.14
BNC + N	42.31	7.37	4.85
BNC + N + V	46.4	6.64	4.86

**Table 3 polymers-13-01995-t003:** Mean values of the material parameters (measurement error 0.6%).

ID	Ultimate Strain	Ultimate Stress, MPa	Young Modulus (Small Deformations)
DM (cadaveric)	3.08	2.58	1.15
DM (fresh)	2.99	2.29	1.27
BNC	1.09	0.58	31.6
BNC + N + V	1.12	0.75	34.37

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
