# Peer review of "The Effect of Adding Modified Chitosan on the Strength Properties of Bacterial Cellulose for Clinical Applications"

_polymers, 2021, doi:10.3390/polym13121995_

Round 1

Reviewer 1 Report

The manuscript (polymers-1225350) entitled "From Thickness Towards Controlling The Mechanical Properties Of Dura Mater Substitutes: Mechanical Experiment And Critical Review" comes under the scope of the journal and found suitable for publication in polymers. Indeed, the manuscript majorly revised for the following suggestions and concerns.

  1. It is suggested to modify the current title of the manuscript. It should be simple, concise, and representing the aim/objective of the investigated research work.
  2.  The abstract should also be modified. It should represent the background, experimental design, results, and conclusion of the investigated research.  The author should also highlight the novelty and significance of the current research work in the revised abstract.
  3. It is suggested to include the finding and outcome of a similar line of research carried out by other researchers in the introduction section and highlight the current gap which is fulfilled through this research investigation.
  4. It is suggested to introduce the purpose of the inclusion of Novochizol,  Vancomycin, and BNC  in the current research investigation in the introduction section.    
  5. It is suggested to reorganize the materials and method section. It should be divided into different sections/sub-section to represent the exact experimental design of the current investigation. 
  6. It is suggested to reorganize the results section into different sections/sub-section to represent the findings and outcome of the current investigation. Results should also be represented in tabular form for easy understanding to the reader.  
  7. Authors are advised to highlight the influencing parameters affecting the mechanical strength and results should be represented in tabular form.
  8. It is suggested to include the future directions for the current research investigation in the conclusion section for the researcher interested in a similar line of research.  Also, highlight the limitations of the current investigation.  

Author Response

First of all, we thank the Reviwer for their attention on our work and inform the following in response to comments:

1. It is suggested to modify the current title of the manuscript. It should be simple, concise, and representing the aim/objective of the investigated research work.

Answer (1): The title has been changed.

2. The abstract should also be modified. It should represent the background, experimental design, results, and conclusion of the investigated research.  The author should also highlight the novelty and significance of the current research work in the revised abstract.

Answer (2): The abstract has been changed.

3. It is suggested to include the finding and outcome of a similar line of research carried out by other researchers in the introduction section and highlight the current gap which is fulfilled through this research investigation.

Answer (3): The section has been added to the introduction.

4. It is suggested to introduce the purpose of the inclusion of Novochizol,  Vancomycin, and BNC  in the current research investigation in the introduction section.    

Answer (4): The section has been added to the introduction.

5. It is suggested to reorganize the materials and method section. It should be divided into different sections/sub-section to represent the exact experimental design of the current investigation. 

Answer (5): Materials and Method section has been split into new subsections.

6. It is suggested to reorganize the results section into different sections/sub-section to represent the findings and outcome of the current investigation. Results should also be represented in tabular form for easy understanding to the reader.  

Answer (6): Results section has been split into new subsections.

7. Authors are advised to highlight the influencing parameters affecting the mechanical strength and results should be represented in tabular form.

Answer (7): The Table A1 was moved into the main text.

8. It is suggested to include the future directions for the current research investigation in the conclusion section for the researcher interested in a similar line of research.  Also, highlight the limitations of the current investigation.  

Answer (8): The limitations were added into the Discussion section. The future directions have been added into the Discussion and Conclusion sections.

Reviewer 2 Report

In the manuscript polymers-1225350, entitled “From thickness towards controlling the mechanical properties of dura mater substitutes: mechanical experiment and critical review”, by authors: Anna Lipovka, Alexei Kharchenko, Andrey Dubovoy, Maksim Fillipenko, Vyacheslav Stupak, Alexander Mayorov, Vladislav Fomenko, and Daniil Parshin, were investigated the mechanical properties of a composite material of bacterial nanocellulose - chitosan nanoparticles - vancomycin to close dura mater defects. They compared data with preserved and native human dura mater. Authors used a different technique of fixing the samples, which increased the area of the contact surface of the clamps and the sample, due to the irregularities of the clamps themselves. After careful reading, there are many important issues that this investigation must reveal in the form and the contents. Furthermore, many things are unclear or not exposed in an exhaustive way and it is needed to make additional improvements:

Abstract:

  • It is stated: ..."R2 = 0.235362 for BNC + Novochizol + Vancomycin, compared to R2 = 0.040461 for native BNC", but this data does not exist in the manuscript results and discussion. It is needed to harmonize data in presented analysis in manuscript text with abstract.

Keywords:

  • "Novochisol" or " Novochizol®"? This is trademark of chitosan nanoparticles, it is better to use chemical name, or generic name, instead trademark name.
  • It will be useful to add vancomycin as keyword.

  1. Introduction
  • What are the other characteristics - specifics of chitosan nanoparticles produced as Novochizol?
  • In my opinion it is better to remove parts of discussion in the introduction part (lines: 202-276 after line 54, and lines: 312-330 after line 67).
  • Figure 1. Authors should cite document, if pictures are previously published.
  • Authors should cite their previously published report at virtual congress, and add at the reference part:

Parshin, A. Lipovka, A. Kharchenko, Investigation of the mechanical properties of a composite material of chitosan-vancomycin-nanocellulose nanoparticles of bacterial origin to close dura mater defects, 11th World Biomaterial Congress, 11-15 December, 2020.  https://www.postersessiononline.eu/173580348_eu/congresos/WBC2020/aula/-WBC2020_3305_WBC2020.

Line 50:  "2. Xenograft (collagen implants: :"...  Please, it is needed to delete duplicated colon.

Line 54: "4. Biopolymers ( Chitosan..." Please, it is needed to delete space between bracket and word.

  • What is the reason for obtaining a new composite of bacterial nanocellulose with Novochizol and vancomycin?
  • Why did the authors choose the vancomycin as drug model? It is needed to write basic information about the drug and the importance of its inclusion in the composite.
  • Word vankomycin is generic name of drug and it must be written in lowercase.

  1. Material and Methods

Line 70:  It should be: 2-5°C

Methods for the synthesis of modified BNC

  • Line 82: Please, it is needed to add precise data about number of registered trademark for used chitosan nanoparticles: "Registered International trademark Novochizol No. 1540749, and in U.S. Patent and Trademark Office No. 6297647".
  • Lines 88-90: It stated: "Samples were individually immersed in a Novochizol solution in a plastic tube (50 ml) at a ratio of sample volume to Novochizol of 1:10."

It is necessary to describe exactly which samples are used. Are these bacterial nanocellulose samples? Please, provide this data.

When was vancomycin added, and in what amount?

Line 93: "Morphological analysis of unimplanted samples..."

  • Please, it is needed to provide necessary data about the applied method for this morphological analysis and device (type, producer, country).
  • Morphological analyze after implantation is required. Additional scanning electron microscopy (SEM) or atomic force microscope (AFM) should be useful for morphological analysis.
  • Did the authors examine structure characterization, fiber and pore sizes, the effect of hydration of new composite material?

2.1. Thickness measurement and material cutting

  • Presented data in Figure 2 should be discussed in the manuscript text, including drawn 1-5 "dog-bone" shapes, not only in figure caption.
  • To compare the obtained results, it would be better to present the analysis of bacterial nanocellulose + vancomycin to observe the effect after addition of chitosan nanoparticles - Novochizole in new composite material.

Mechanical test protocol

  • Authors should cite their previously published report at virtual congress in the title and text analysis for Figures 3 (correspond to fig. 1 and 2), 6 (correspond to fig. 3), 7 and 8 (correspond to fig. 4), because they seem identical.
  • Please, it is needed to provide necessary data in the manuscript text for the applied method and analysis for tensile test, presented on Figure 4.(a).
  • It is necessary to define all abbreviation (BCN, BNC+C ?, DM) presented in the graph in Figure 4.(b), harmonize, and add in the figure

  1. Results
  • Line 163: Authors should clearly describe and recalculate total number of carried out experiments, and also harmonize it in the Abstract.
  • In Figure 5 title the parts: "On the left" and "On the right" should be deleted. Analysis of presented results in this figure should be near the part of Statistical analysis, and please reorder it as figure 8.
  • It is needed to define or correct the acronym BNC+N in Figure 6 (It should be figure 5 after reordering). It is necessary to synchronize presented title and results in 6.b with title and analysis of graph in figure 3 in the previously published report at virtual congress: "Fig. 3. (above) The technique of attaching marks to the specimen for the extensometer...". Also, what are the characteristics of the Tutopatch material mentioned in the title of Figure 6? It is necessary to include it in manuscript discussion and analysis.
  • Figure 7. "Linear regression for BNC (a) and BNC+Novochizol R+Vancomycin (b) samples for stress-strain relationship" correspond to the part of figure 4 in the previously published report at virtual congress: "... Linear regression for BNC (3) and BNC + Novochisol (4) samples for stress-strain relationship." It is necessary to synchronize presented results and discussion. (Figure 7. should be figure 6 after reordering.)
  • Figure 8. " Linear regression for BNC (a) and BNC+Novochizol R+Vancomycin (b) samples for stress-thickness relationship..." correspond to the part of figure 4 in the previously published report at virtual congress: "Linear regression for BNC (1) and BNC + Novochisol (2) samples for stressthickness relationship..." It is necessary to synchronize presented results and analysis. (Figure 8. should be figure 7 after reorder.)

Elemental analysis of Novochizol®

Line 197: It stated: "The following element ratio was established: C 39.73%, H 7.11%, N 7.14%". Please, indicate for which sample these data are displayed. Is this analyze of chitosan nanoparticles Novochizol®?

Line 200: It stated: "Elemental analysis of BNC+N+V sample shows that Novochizol® and bacterial  cellulose are in approximately equal weight proportions."

  • Where is the data for BNC+N+V? Please, it will be better to provide necessary data about elemental analysis for all analyzed samples in the table.
  • The structural formulas of the obtained composite material would be useful for better representation.

Additional analysis

  • Did the authors examine the efficacy of vancomycin incorporation and release kinetic of vancomycin from composite samples BNC with and without chitosan nanoparticles?
  • Also, did the authors analyze the antibacterial and antifungal properties of the composite?

  • Please, it is needed to rewrite sentences in 3rd person plural and passive voice and avoid 1st person plural (Lines 7, 9, 66, 106, 137, 138, 140, 148, 149, 151, 160, 173, 186, 249, 297, 303, 316, 327, 328, 337, 340).
  • It will be enough presentation of decimal numbers with 3 to 4 digits (lines: 177, 183, 193).

Lines 346-349: Please, it is needed to add application numbers, titles and priority dates for a patent for a method for measuring the mechanical properties of BNC, as well as for a patent for a utility model for biomaterials and biopolymers fixation.

Abbreviations

  • Synchronize all used abbreviations, especially abbreviation BNC + N + V with a display in the text: BNC + N and BNC + C.

Table A1 should be included in the main text and fully analyzed in discussion part.

References

  • It is necessary to add journal titles to References, especially: 3 (Journal of Neurological Surgery Part B), 8 (Materials Science and Engineering: C), 9 (British Journal of Neurosurgery), 13 (Anatomic Bases of Medical, Radiological and Surgical Techniques), 17 (Indian Journal of Neurotrauma).
  • The part of journal name in the reference 14. is needed to be deleted: "An Official Journal of The Society for Biomaterials, The Japanese Society for Biomaterials, and The Australian Society for Biomaterials and the Korean Society for Biomaterials".
  • Line 450: reference 32: It is needed to capitalize the journal name!
  • The title of manuscript should be changed after revision.
  • According to all these considerations and others which are not highlighted, the manuscript in the present form is not suitable for publication in Polymers journal.
  • I suggest to the authors to revise and improve this manuscript according to the presented remarks and to resubmit it.

Best regards,

Reviewer

Round 2

Reviewer 1 Report

The revised manuscript (polymers-1225350) improved well and incorporated all given suggestions nicely. The present version of the manuscript is well organized and should be considered for publication in its current form in Polymers journal.

Author Response

The authors thank the reviewer for the attention to the manuscript.

Reviewer 2 Report

Authors provided corrections and improvement in the revised manuscript polymers-1225350, entitled “The effect of adding modified chitosan on the strength properties of bacterial cellulose for clinical applications. After careful reading, there are some additional improvements before publication: 

Lines 262-263: The sentences: "Semisoft AFM probes NSG01 (NT-MDT, Russia) with resonant frequency 138 kHz and nominal tip radius 10 nm were used for the studies with the optimized setpoint force. All recordings were done with a scan rate of 0.5 Hz and resolution 512x512 points." should be included in the part: 2.5. Atomic Force Microscopy.

422-423: Conflicts of Interest: V.V. Fomenko is the inventor of patent covering Novochizol™ technology and uses and owns stock in Novochizol SA. Please, it is needed to add application number, and title for this patent.

Author Response

The authors thank the reviewer for the attention to the manuscript. All suggestions have been taken into account.